# Performance versus resilience in modern quark-gluon tagging

Anja Butter[1,2], Barry M. Dillon[1], Tilman Plehn[1] and Lorenz Vogel[1]

**1** Institut für Theoretische Physik, Universität Heidelberg, Germany
**2** LPNHE, Sorbonne Université, Université Paris Cité, CNRS/IN2P3, Paris, France

## Abstract

**Discriminating quark-like from gluon-like jets is, in many ways, a key challenge for many LHC analyses. First, we use a known difference in PYTHIA and HERWIG simulations to show how decorrelated taggers would break down when the most distinctive feature is aligned with theory uncertainties. We propose conditional training on interpolated samples, combined with a controlled Bayesian network, as a more resilient framework. The interpolation parameter can be used to optimize the training evaluated on a calibration dataset, and to test the stability of this optimization. The interpolated training might also be useful to track generalization errors when training networks on simulations.**

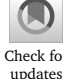

## 1 Introduction

Jets are the main analysis objects at the LHC, and the success of the LHC program is, to a large degree, being driven by an improved understanding of jets experimentally and theoretically. In practice, the main task in jet physics is to predict their features precisely, and to use them to tag the parton initiating the jet. The improved understanding of subjet physics at the LHC has allowed us to skip high-level observables and instead analyze jets using low-level detector

output employing modern machine learning [1]. ML-tools can incorporate all available information in a jet and significantly improve the performance of classic, multivariate jet taggers. While, in the interest of optimality, taggers should be trained on data, ATLAS and CMS follow a more conservative approach and train taggers on simulations. This attitude reflects a flattering trust in theoretical simulations, but it also creates new sources of uncertainties.

In this paper we add uncertainty-aware features to the CMS ParticleNet tagger [2], using a Bayesian classification network setup [3], and propose an interpolated training method using conditional networks. This combination allows us to capture different sources of uncertainties [4] while protecting the performance of the tagger. The Bayesian setup would raise a flag when the training datasets are too inconsistent to be combined. The conflict between optimal performance and uncertainty control is the key weakness of adversarial training approaches and makes it preferable to instead use nuisance parameters to describe systematics [5]. For theory uncertainties, adversarial approaches are not even likely to cover the full uncertainty range [6].

ML-methods for jet tagging [7] can be applied to the whole range of top jets, Higgs jets, $W/Z$-jets, $\tau$-jets, bottom or charm jets, all the way to quark vs. gluon jets. From a theoretical and an experimental perspective it is easiest to tag partons of which only the decay products hadronize. For instance top taggers then look for distinctive features like the jet mass or the multiplicity of subjet constituents [8–10]. Being a well-defined problem, top tagging has played a key role in developing and establishing a wide range of network architectures [11], including uncertainty-aware extensions.

In contrast, quark vs. gluon tagging is not actually defined theoretically, beyond leading-order in QCD and including parton splittings. Still, because of its great analysis potential, quark–gluon tagging has a long history [12–19], including early applications at the LHC [20–25]. In spite of the serious theoretical challenges [26–35], efficient ML-approaches have been devised to separate "quark jets" from "gluon jets" [36–41]. They include study of hadronization and detector effects [42] and modern network architecture like transformers [43], Lorentz-equivariant networks [44], and normalizing flows [45]. One way to overcome the fundamental problem of defining quark and gluon jets is to instead use well-defined hypotheses in terms of LHC signatures, for instance mostly quarks in LHC signals like weak boson fusion vs. gluons in QCD backgrounds [46–50]. An alternative method is to train a classifier without labels, just on samples with an enhanced partonic quark or gluon fraction [51].

In this paper we first look at a known issue, namely the differences between HERWIG and PYTHIA jets and the effect of these differences on ML-taggers introduced in Sec. 2.1. To control the cutting-edge ParticleNet tagger and understand its output better, we present its Bayesian variant in Sec. 2.2. It allows us to understand the problem of quark–gluon taggers trained on HERWIG and PYTHIA, as shown in Fig. 1, and makes it obvious that a naive resilience improvement through decorrelation will massively hurt the performance of the tagger, as discussed in Sec. 3. In Sec. 4 we target this problem through a new, interpolated training of the conditional ParticleNet tagger on two distinct samples. We realize this interpolation with the same ParticleNet classifier. After discussing this method in detail, we extend it to a fresh look at a more interpretable, continuous calibration of jet taggers.

## 2 Dataset and classification network

One of the most exciting goals of subjet tagging is the discrimination of quarks versus gluons. The precise task is not well-defined beyond leading order in QCD, but it approximates important questions like how to identify electroweak decay jets or how to separate weak boson fusion from QCD backgrounds. In both cases, the signals are quark-enriched, while most QCD

jets at the LHC come from gluon emission. Another aspect which makes quark–gluon tagging especially interesting is that there exists a study which raises questions about the behavior of ML-taggers in this application.

## 2.1 Quark–gluon datasets

The starting point of our study are two datasets of simulated quark and gluon jets [52–54], each with 2M jets, one generated with PYTHIA and one generated with HERWIG. The two samples are generated using the partonic processes

$$q\bar{q} \to Z(\to \nu\bar{\nu}) + g\,, \qquad \text{and} \qquad qg \to Z(\to \nu\bar{\nu}) + (uds)\,, \tag{1}$$

at the 14 TeV LHC, simulated with PYTHIA 8.226 [55, 56] and with HERWIG 7.1.4 [57]. Both setups use default tunes and shower parameters. Hadronization and multi-parton interactions (MPI) are turned on, and we do not consider jets with charm or bottom quark content (at the level of the hard process). The jets are defined through anti-$k_T$ algorithm [58, 59] in FASTJET 3.3.0 [60] with a radius of $R = 0.4$. No detector simulation is included, which cuts into the realism of the analysis, but allows us to extract the underlying question and issues and solve them before adding detector simulations to the problem. For each event the dataset keeps the leading jet, provided

$$p_{T,\text{jet}} = 500\dots 550\,\text{GeV}\,, \qquad \text{and} \qquad |\eta_{\text{jet}}| < 1.7\,. \tag{2}$$

If we assume that all light-flavor jet constituents are approximately massless, each jet $x_i$ is defined by

$$x_i = \left\{ (p_T, \eta, \phi)_k \right\}\,, \qquad \text{with} \qquad k = 1, \dots, n_C\,. \tag{3}$$

For our analysis we allow for up to $n_C = 100$ constituents per jet. The jets are zero-padded with constituents, and all constituents have azimuthal angles $\phi$ within $\pi$ of the jet.

We refer to the final-state jets from the two partonic processes in Eq. (1) as quark and gluon jets, even through it is clear that this statement is scale dependent and only defined at leading order in perturbation theory. A more appropriate way of referring to these jets would be in the sense of semi-supervised learning and quark-enhanced vs. gluon-enhanced samples. A standard way of realizing this setup would be jets reconstructed as coming from a two-body $Z$-decay vs. jets produced in association with a Higgs boson.

We supplement the PYTHIA and HERWIG datasets with a third simulation of the two processes in Eq. (1) using SHERPA 2.2.10 [61], again with the default tune and shower parameters. Using PYHEPMC [62, 63], a PYTHON wrapper for the HEPMC2 [64] library, we select the constituent coordinates of those final-state particles not labelled as neutrinos. The SHERPA jets are defined through the PYJET [62, 65] interface to FASTJET. From a physics perspective, the SHERPA jet resembles the HERWIG jets through the common use of cluster fragmentation, but we will see that the numerical results differ.

Each of our three jet datasets consists of 20 files with 100k jets each, equally split between quark and gluon jets. For each generator we divide the dataset into training/validation/test subsets with 200k/50k/50k jets for quarks and gluons, each, unless mentioned otherwise.

For state-of-the-art jet tagging we need to include particle identification (PID) information. Our PYTHIA and HERWIG datasets include two forms of PIDs [52], (i) the full particle-ID information from PYTHIA or HERWIG, and (ii) experimentally realistic particle IDs. We follow the ParticleNet approach [2], using the five particle types electron, muon, charged hadron, neutral hadron, and photon, plus the electric charge as input to the network. The standard encoding by the Particle Data Group in terms of large and irregular integer values is not an ideal ML-input. Instead, we use a one-hot encoding of our experimentally realistic PIDs.

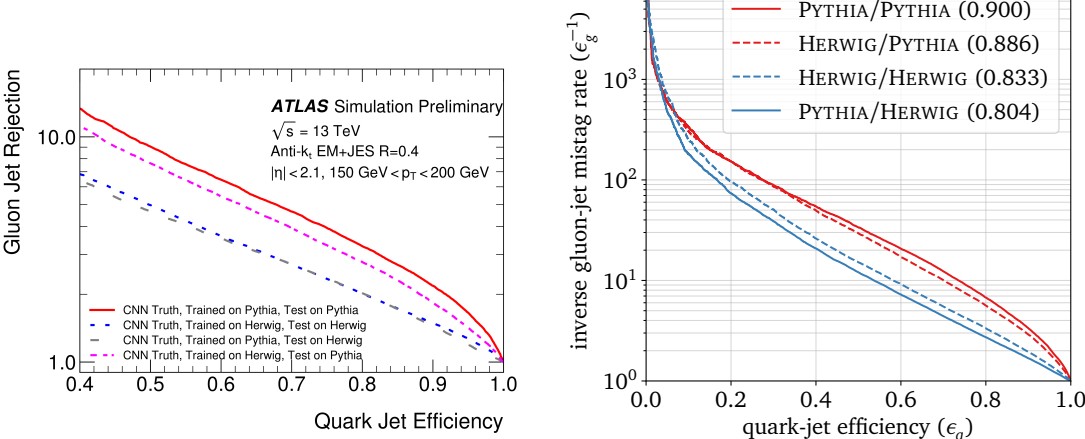

Figure 1: *Left:* preliminary result of the ATLAS study on quark–gluon tagging, raising questions on the best way to train such a tagger. Figure from Ref. [22]. The same pattern has been observed in Fig. 8 of Ref. [37]. *Right:* our results on the same task, using a new, Bayesian version of ParticleNet-Lite [2], trained on 400k PYTHIA and HERWIG jets each, with the parameters given in Tab. 2. The first generator in our labeling always refers to the training dataset and the second one to the test dataset. The values in parantheses indicate the respective area under curve (AUC).

**High-level observables**

There exist standard kinematic observables for subjet physics, specifically quark–gluon discrimination [48], for instance the multiplicity of constituents or particle flow objects ($n_{\mathrm{PF}}$), the radiation distribution or girth ($w_{\mathrm{PF}}$) [66, 67], the width of the $p_{\mathrm{T}}$-distribution of the constituents ($p_{\mathrm{T}}D$) [23], or the weighted angular correlator ($C_{0.2}$) [68]. They are defined in terms of the jet constituents as

$$
n_{\mathrm{PF}} = \sum_i 1\,, \qquad\qquad w_{\mathrm{PF}} = \frac{\sum_i p_{\mathrm{T},i} \Delta R_{i,\mathrm{jet}}}{\sum_i p_{\mathrm{T},i}}\,,
$$
$$
p_{\mathrm{T}}D = \frac{\sqrt{\sum_i p_{\mathrm{T},i}^2}}{\sum_i p_{\mathrm{T},i}}\,, \qquad\qquad C_{0.2} = \frac{\sum_{i,j} p_{\mathrm{T},i} p_{\mathrm{T},j} (\Delta R_{ij})^{0.2}}{\left(\sum_i p_{\mathrm{T},i}\right)^2}\,. \tag{4}
$$

Distinguishing quark jets from gluon jets exploits two features encoded in these observables [42, 69]. First, the QCD color factors for quarks are smaller than for gluons, which means radiating a gluon off a hard gluon versus off a hard quark comes with the ratio $C_A/C_F = 9/4$. This leads for a higher multiplicity and broader girth for hard gluons. Second, the quark and gluon splitting functions differ in the soft limit. The harder fragmentation for quarks leads to quark jet constituents carrying a larger average fraction of the jet energy, tracked by $p_{\mathrm{T}}D$.

In Fig. 2 we show these four distributions for the quark and gluon jets simulated by PYTHIA, SHERPA, and HERWIG. The biggest difference appears in $n_{\mathrm{PF}}$, where the quark distributions from the three generators are similar, but the gluon distributions vary significantly. The maximum of the broad peak is the smallest, $n_{\mathrm{PF}} \sim 40$ for the HERWIG gluons and the largest, $n_{\mathrm{PF}} \sim 45 \ldots 50$ for PYTHIA gluons. This difference between HERWIG and PYTHIA jets is not cause by noise or a broader distribution, but by an actually different prediction for $n_{\mathrm{PF}}$.

This difference in $n_{\mathrm{PF}}$ vanishes for $w_{\mathrm{PF}}$, indicating that it comes from infrared and collinear unsafe regions of phase space, and might become less relevant once we include detector effects. We emphasize that this does not mean we should expect the shower algorithms to fail,

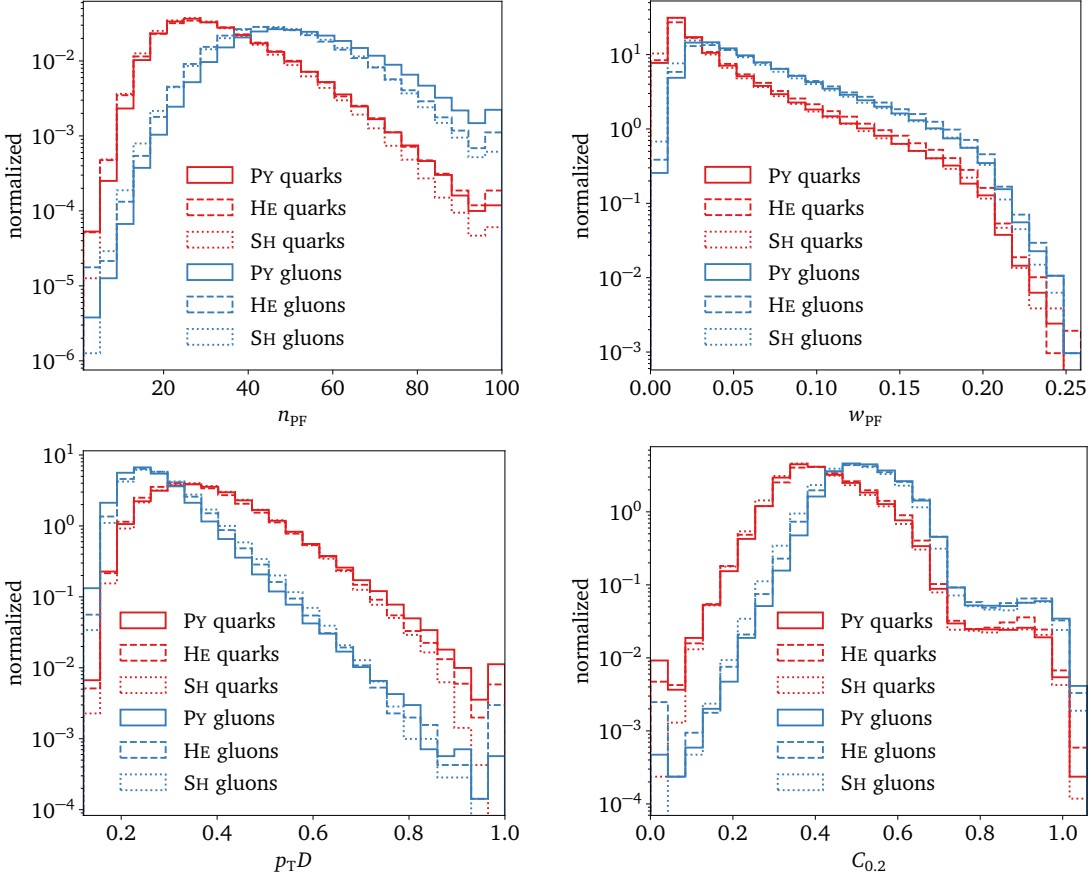

Figure 2: High-level observables, as defined in Eq. (4), for the three different generators, split into quark and gluon jets.

but that these difference are not easily computable in perturbative QCD. Similarly, the $p_T D$ distributions are significantly different for quarks and gluons, combined with a small shift in the position of the comparably sharp gluon peaks from the different generators. Finally, the only actual two-constituent correlation $C_{0.2}$ is also different for quarks and gluons, but consistent for the different generators. We have studied a range of additional high-level operators and traced significant deviations between the gluon jets from the different generators to a strong correlation with $n_{PF}$.

In Fig. 3 we also show the correlations between the same observables, for each of the three generators and separated into true quark and gluon jets. All observables are correlated with the most powerful $n_{PF}$, but this correlation is not very different for quarks and for gluons, suggesting that a multi-dimensional analysis will be dominated by the completely understood shifts in $n_{PF}$.

Table 1: First Wasserstein distance, or earth mover's distance, between quark and gluon distributions for the observables defined in Eq. (4). We show 200k quark jets and 200k gluon jets for each generator.

| generator | $n_{PF}$ | $w_{PF}$ | $p_T D$ | $C_{0.2}$ |
|---|---|---|---|---|
| PYTHIA | 2050 | 3207 | 4000 | 1316 |
| SHERPA | 1711 | 3149 | 3217 | 1112 |
| HERWIG | 1326 | 2910 | 3406 | 1128 |



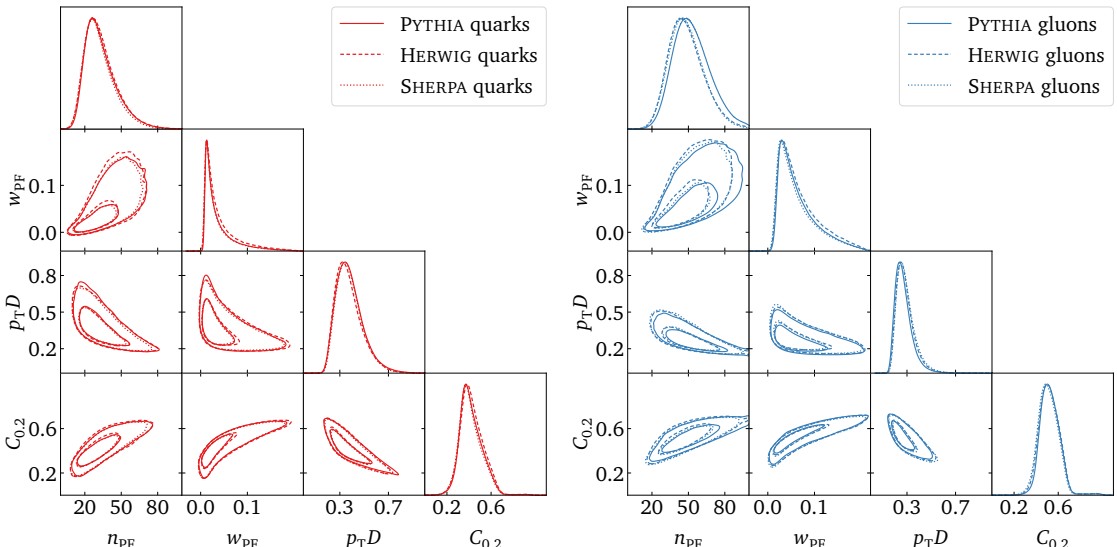

Figure 3: Correlations between the high-level observables from Eq. (4). We show results for the three different generators, split into quark jets (*left*) and gluon jets (*right*).

To judge the relevance of the difference in $n_{PF}$ from the different generators for quark–gluon tagging we can separate quarks from gluons based on the individual observables given in Eq. (4). We can estimate the power of the individual distributions using the Wasserstein distances between 200k quark and gluon jet histograms, as given in Tab. 1. For all observables the PYTHIA jets are most easily separated, followed by SHERPA for $n_{PF}$ and $w_{PF}$, whereas HERWIG predicts a stronger discrimination power for $p_T D$ than SHERPA. The actual values of the Wasserstein distance for the different kinematic observables depends on the detailed shape and does not correlate with the separating power of a kinematic cut. We show the corresponding ROC curves in Fig. 4, generated by choosing such a cut value for each observable. The $n_{PF}$-based and $p_T D$-based tagging shows a significant degradation when tagging HERWIG jets as compared to the easier-to-separate PYTHIA quarks and gluons. This confirms the observation from Fig. 2, where both distributions for HERWIG gluons are further from the common quark distributions than they are for the PYTHIA gluons. In contrast, the tagging performance from $w_{PF}$ and $C_{0.2}$ is unaffected by the choice of simulation and in general also much weaker.

To summarize the key result from this simple study — the most powerful observables for quark–gluon tagging show a significant shift in the gluon predictions between HERWIG and PYTHIA. This shift brings the HERWIG gluons closer to quarks.

## 2.2 Bayesian ParticleNet

To work with a controlled cutting-edge ML-tagger we develop a Bayesian version of the ParticleNet(-Lite) graph convolutional network architecture [2] adapted from TENSORFLOW to PYTORCH, to be able to use our standard Bayesian network. For a detailed discussion of Bayesian networks we refer to some original Bayesian network papers [70–72] and the didactic introduction in Ref. [1]. The ADAMW optimizer [73, 74], with a weight decay of $10^{-4}$, minimizes the usual binary cross-entropy loss combined with a sigmoid activation function for the classification task,

$$\mathcal{L}_{PN} = -\frac{1}{M} \sum_{i=1}^{M} y_i \log f(x_i) + (1 - y_i) \log\big(1 - f(x_i)\big), \qquad (5)$$

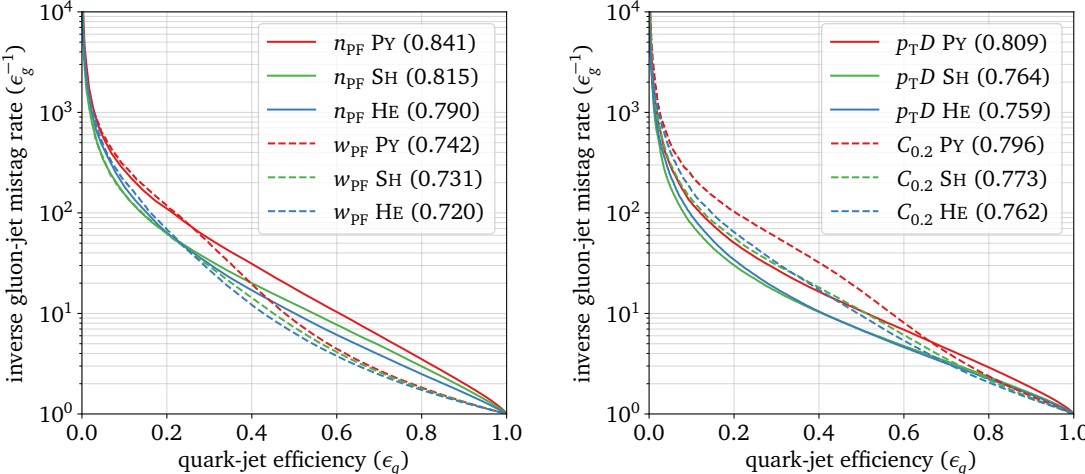

Figure 4: AUCs for quark–gluon discrimination based on individual high-level observables of Eq. (4), simulated with PYTHIA, SHERPA and HERWIG.

where $M$ is the mini-batch size, $f(x_i) \in [0, 1]$ the model prediction for jet $i$, and $y_i \in \{0, 1\}$ the jet truth-label. The two term in the loss lead to a classification of

$$
\begin{aligned}
f(x_i) &\to y_i = 1, & \text{quark signal}, \\
f(x_i) &\to y_i = 0, & \text{gluon background}.
\end{aligned}
\tag{6}
$$

We adopt the learning-rate scheduling from Ref. [2]. The feature input to the ParticleNet are the hardest 100 jet-constituent particles, specifically

$$
\left\{ \Delta\eta_k, \ \Delta\phi_k, \ \Delta R_k, \ \log p_{\mathrm{T},k}, \ \log \frac{p_{\mathrm{T},k}}{p_{\mathrm{T,jet}}}, \ \log E_k, \ \log \frac{E_k}{E_{\mathrm{jet}}}, \ \mathrm{PID}_k \right\},
\tag{7}
$$

where the first coordinates are computed relative to the jet axis. The distance in $\Delta\eta_k$ and $\Delta\phi_k$ are used to compute the distances between particles in the first edge convolution (EdgeConv) block (coordinate input). The PID information includes the particle charge [2, 52].

While deterministic neural networks adapt a large number of weights to approximate a training function, Bayesian neural networks (BNNs) learn distributions of these weights [72].[1] We can then sample over the weight distributions to produce a central value and an uncertainty distribution for the network output. In LHC physics, Bayesian networks can be applied to classification [3], regression [75, 76], and generative networks [4, 77, 78]. While it is in general possible to separate these uncertainties into statistical and systematic (stochasticity [75] or model limitations [76]), we know that our number of training jets is sufficiently large to only leave us with systematic uncertainties from the training process.

The Bayesian loss follows from a variational approximation of the conditional probability for the network parameters. It combines a likelihood loss with a regularization through a prior for the weight distributions,

$$
\mathcal{L}_{\mathrm{BPN}} = -\frac{1}{M} \sum_{i=1}^{M} \log p(y_i | x_i, \omega) + \frac{1}{N} \mathrm{KL}\big[ q_{\mu,\sigma}(\omega), p_{\mu,\sigma}(\omega) \big]
\tag{8a}
$$

$$
\approx -\frac{1}{M} \sum_{i=1}^{M} \log p(y_i | x_i, \omega) + \frac{1}{2N} \sum_{\text{weights } \omega_j} \big( \mu_j^2 + \sigma_j^2 - \log \sigma_j^2 - 1 \big),
\tag{8b}
$$

---

[1]In that sense there is nothing Bayesian about BNNs, they can just be viewed as an extremely efficient way to train network ensembles.

Table 2: Bayesian ParticleNet-Lite (BPN-Lite) architecture and hyperparameters [2].

| hyperparameter | BPN-Lite architecture |
|---|---|
| number of EdgeConv blocks | 2 |
| number of nearest neighbors | 7 |
| number of channels for each EdgeConv block | $(32, 32, 32), (64, 64, 64)$ |
| channel-wise pooling | average |
| fully-connected layer | 128 and ReLU |
| dropout probability | 0.1 |
| number of epochs | 100 |
| batch size | 128 |
| number of constituents | 100, with highest $p_T$ |
| training/validation/testing | 400k/100k/100k |
| signal-to-background ratio | 1.0 |
| re-sampling for testing | 80 times |

where we choose the prior as a Gaussian with mean zero and width one and use the fact that the resulting weight distributions will become approximately Gaussian as well, described by $\mu_j$ and $\sigma_j$. A change of prior has been shown to not affect the network output [3]. As in Eq. (5) $M$ denotes the mini-batch size, and $N$ is the number of training jets.

The parameters $\mu_j$ and $\sigma_j$ define the model parameters $\omega_j$ of the Bayesian network and need to be trained. In our case, only the weights in the linear and 2D-convolutional layers are extended to Gaussian distributions. The hyperparameters of the original ParticleNet(-Lite) network and its Bayesian counterparts are given in Tab. 2. We use the same BPN-Lite network for quark vs. gluon discrimination and for the generator reweighting which we will introduce in Sec. 4.

The performance of the BPN-Lite quark–gluon classifier is illustrated in the right panel of Fig. 1. Note that the first generator in our labeling always refers to the training dataset and the second one to the test dataset. Independent of the competitive AUC values we see that, as before, the network trained and tested on PYTHIA performs best, closely followed by the network trained on HERWIG and tested on PYTHIA. This suggests that the choice of training sample only has a small effect. In contrast, when we test networks on HERWIG the performance drops significantly, with the consistent training on HERWIG superseding the training on the alternative PYTHIA dataset in terms of AUC and background-rejection performance. This hierarchy indicates that, indeed, PYTHIA quarks and gluons are easier to separate than the HERWIG quarks and gluons, and that the key features for this classification are similar for the two generators. We will study this aspect more closely in the following section.

## 3 Where have all the gluons gone?

Trying to solve the puzzle of quark–gluon taggers trained and tested on different generators will lead us to the more general question, namely how to control classification networks trained on one dataset and tested on another. All combinations of training and testing the BPN-Lite tagger are illustrated in Fig. 5, with some of the main results collected in the two tables. Since the original PYTHIA and HERWIG results form the two extreme poles, we assign the three

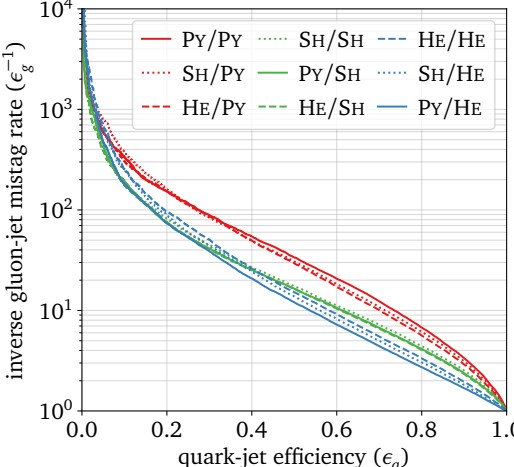

| AUC | | training | | |
| --- | --- | --- | --- | --- |
| | | PYTHIA | SHERPA | HERWIG |
| testing | PYTHIA | 0.900 | 0.893 | 0.886 |
| | SHERPA | 0.856 | 0.863 | 0.853 |
| | HERWIG | 0.804 | 0.820 | 0.833 |
| $\epsilon_g^{-1}(\epsilon_q=0.50)$ | | training | | |
| | | PYTHIA | SHERPA | HERWIG |
| testing | PYTHIA | 33.8 | 30.8 | 29.4 |
| | SHERPA | 16.5 | 17.4 | 16.4 |
| | HERWIG | 12.1 | 13.8 | 15.2 |

Figure 5: *Left:* ROC curves for training and testing on PYTHIA, SHERPA, and HER-WIG in different combinations. *Right:* AUC and background-rejection performance of the BPN-Lite for quark–gluon tagging, trained and tested on the three different generators.

datasets to the real-world problem of training a tagger on two independent training datasets and testing it on independent data as

- labelled training dataset 1: PYTHIA,

- labelled training dataset 2: HERWIG,

- independent test dataset: SHERPA.

We will start by comparing different trainings on the labelled PYTHIA and HERWIG datasets, as motivated by Fig. 1 and eventually add SHERPA results as an independent test, in the sense of actual data analyzed by the tagger.

**Performance comparison**

The Bayesian nature of the BPN-Lite tagger comes with two pieces of information, which allow us to understand the network training. First, the Bayesian tagger provides a per-jet uncertainty $\sigma_{\text{pred}}(x_i)$ on the classification output $\mu_{\text{pred}}(x_i) \in [0, 1]$. This means we can separate jets for which the network training leads to a confident classification from jets where the training provides less information. Second, the final sigmoid layer of the classification network leads to a correlation of $\mu_{\text{pred}}$ and $\sigma_{\text{pred}}$, namely

$$\sigma_{\text{pred}}(x_i) \propto \mu_{\text{pred}}(x_i)\big[1-\mu_{\text{pred}}(x_i)\big]. \tag{9}$$

This inverse parabola correlation is a feature of the network structure and has to be present in the Bayesian tagging output, its absence points to a stability issue in the networks training. In Fig. 6 we show the $\mu_{\text{pred}}$- and $\sigma_{\text{pred}}$-distributions for PYTHIA and HERWIG test datasets, after consistently training the networks on PYTHIA and HERWIG. Already the $\mu_{\text{pred}}$-distributions shows three major issues:

1. While the tagging of quarks vs. gluons is never symmetric, training and testing on PYTHIA indicates some gluons confidently identified as gluons $\mu_{\text{pred}}(x_i) \to 0$.

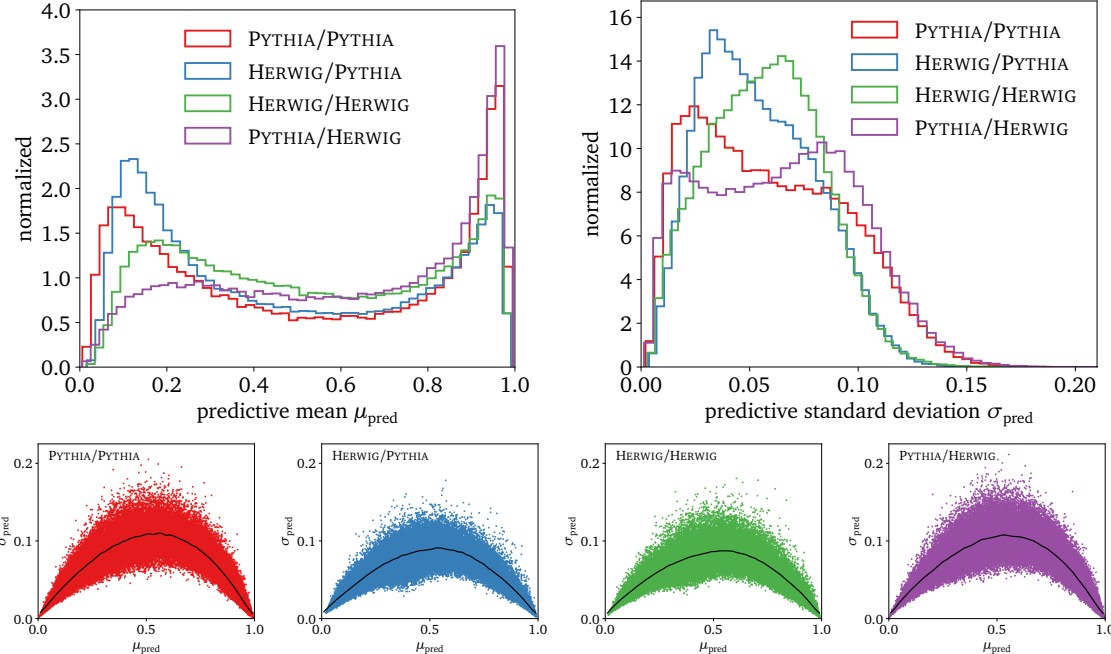

Figure 6: Predictive means ($\mu_{\text{pred}} = 0$ for gluons, $\mu_{\text{pred}} = 1$ for quarks) and standard deviations from the BPN-Lite tagger trained and tested on PYTHIA and HERWIG in different combinations. The lower panels illustrate a stochastic pattern around the correlation of Eq. (9).

2. Training and testing on HERWIG hardly ever allows the network to confidently identify gluons with $\mu_{\text{pred}} \lesssim 0.1$.

3. Training on PYTHIA and testing on HERWIG identifies at least some gluons with as small $\mu_{\text{pred}}$ as training and testing on PYTHIA.

Looking at the $\sigma_{\text{pred}}$-distribution, the results from training on PYTHIA look as expected as long as we test on PYTHIA jets, but tested on HERWIG a slight shoulder around $\sigma_{\text{pred}} \sim 0.07$ develops into a second peak. This peak corresponds to jets or phase-space configurations where the PYTHIA training does not allow for a confident application to HERWIG jets. Second, the general uncertainty after training on HERWIG jets peaks at larger $\sigma_{\text{pred}}$, indicating that the network faces difficulties to extract the relevant features for the tagging, but also drops off at smaller $\sigma_{\text{pred}}$ values than the PYTHIA trainings. This reflects the problem with the single main feature $n_{\text{PF}}$, as expected from our discussion in Sec. 2.1.

Finally, the four lower panels in Fig. 6 show the per-jet correlation of the predictive means and standard deviations. Again confirming our suspicions from Sec. 2.1 that training on HERWIG jets is not completely stable, leading to slight irregularities of the scattering pattern around the inverse parabola predicted by Eq. (9).

**High-level observables**

We can trace back the problems with the performance and stability of the HERWIG training to the high-level observables of Eq. (4). In Fig. 7 we show two of the most interesting kinematic variables in slices of $\mu_{\text{pred}}$, the probabilistic output of BPN-Lite. We know already that $n_{\text{PF}}$ is the leading discriminating feature separating quarks from gluons, while $C_{0.2}$ is the only actual correlator amongst the standard high-level observables. In the upper panels we show PYTHIA jets, in the lower panels HERWIG jets. The slices are bases on consistent training and testing

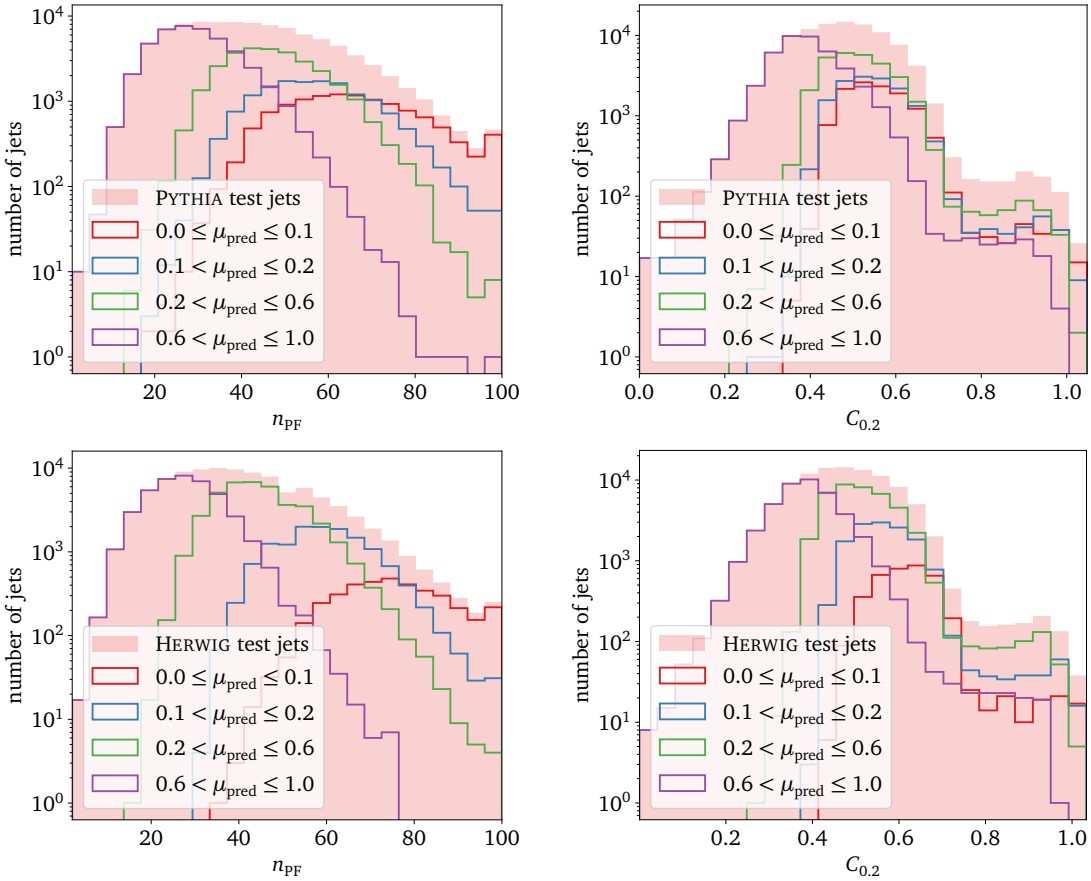

Figure 7: Kinematic distributions defined in Eq. (4). We train the BPN-Lite tagger consistently trained and tested on PYTHIA (*upper*) and on HERWIG (*lower*). The histograms are normalized such that they reflect the fractions of jets in the respective slices in $\mu_{\text{pred}}$, extracted from consistent testing.

on the two samples. For $\mu_{\text{pred}} > 0.6$ the two distributions agree, as expected for correctly identified quarks.

While the two $n_{\text{PF}}$-distributions are very similar for correctly identified quark-like jets with $\mu_{\text{pred}} > 0.6$, differences appear towards the gluon regime and become quite dramatic for correctly identified gluons with $\mu_{\text{pred}} < 0.1$. Requiring increasingly small $\mu_{\text{pred}}$ values for more and more confidently identified gluons, the fraction of jets remaining in these slices from the PYTHIA sample is much larger than it is for the HERWIG sample. While for PYTHIA jets values $n_{\text{PF}} > 60$ indicate confidently identified gluons, HERWIG gluons are harder to identify and typically require $n_{\text{PF}} > 70$ to lead to the rare occurrence of $\mu_{\text{pred}} < 0.1$. In the right panels we show the correlator $C_{0.2}$. While the main difference is the number of jets in the individual slices, we also see that the secondary maximum around $C_{0.2} > 0.8$ is predominantly, but not exclusively populated by gluon jets.

**Predictive uncertainties**

Finally, we can see what the predictive uncertainties tell us in addition to this information from the network performance. For a given tagger the predictive mean $\mu_{\text{pred}}$ and the predictive standard deviation $\sigma_{\text{pred}}$ are strongly correlated through Eq. (9), but this argument does not hold for different training datasets. In Fig. 8 we show the predictive uncertainties the BPN-Lite tagger extracts when training and testing on all possible combinations of PYTHIA, HERWIG, and

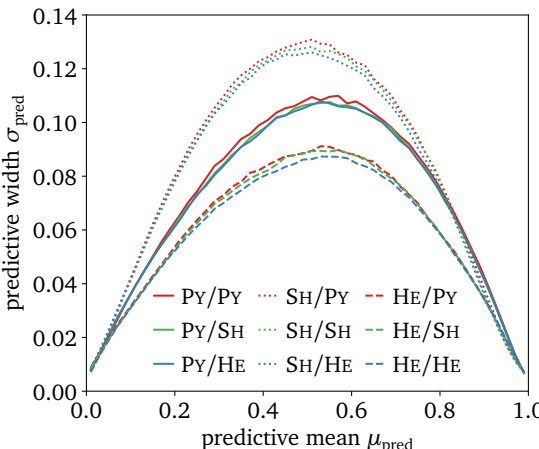

Figure 8: Correlation between the predictive mean and average uncertainty from the BPN-Lite tagger for different combinations of training and testing data.

SHERPA. In the ranges $\mu_{\text{pred}} \sim 0.1 \ldots 0.9$ the different training samples define the size of the predictive uncertainty.

The ranking of the three generators providing the training dataset is independent of the test sample. This confirms that the predictive uncertainty of the Bayesian network reflects almost entirely limitations in the training data. While $\mu_{\text{pred}}$ and $\sigma_{\text{pred}}$ are correlated for a given training dataset, the $\sigma_{\text{pred}}$ values in a given range of $\mu_{\text{pred}}$ are not correlated with the respective $\mu_{\text{pred}}$ values for different generators. For instance, the poorly performing HERWIG training might not exploit features optimally, but it is less affected for instance by the stochasticity of the training data. We also see that any kind of training on PYTHIA and HERWIG will provide smaller uncertainties on the independent SHERPA data than a Bayesian network trained on SHERPA and tested on SHERPA. We again emphasize that this kind of behavior should not appear for $\mu_{\text{pred}}$, because consistent training should provide better performance than inconsistent training, but it can happen for $\sigma_{\text{pred}}$, as it reflects limitations of the training dataset only.

## 4 Resilient interpolated training

Once we have understood what the physics issues and the ML-implications with the PYTHIA and HERWIG training datasets are, we can follow the setup from the beginning of Sec. 3 and see how to best deal with two significantly different training datasets, when the tasks is to identify quarks in a third, independent dataset (SHERPA). This corresponds to the standard ATLAS and CMS strategy, which is to train ML-classifiers on Monte Carlo simulations, understand their behavior, and then apply them to data. The major drawback of this strategy is a generalization error whenever simulations do not reproduce data perfectly. Such a generalization error can introduce a bias, but at the very least it is leading to non-optimal performance. A re-calibration should remove biases, but it will not improve poorly trained taggers. We propose a flexible choice of training data, defining an optimal training dataset by evaluating the tagger performance on an independent calibration dataset.

A related question is how to estimate systematic uncertainties related to the choice of training data. In general, whenever uncertainties can be described reliably, it is preferable to include the corresponding nuisance parameters in the analysis, instead of removing a model dependence through adversarial training [5]. Decorrelating theory uncertainties induced by different datasets is especially tricky, since it enforces an insensitive direction in feature space

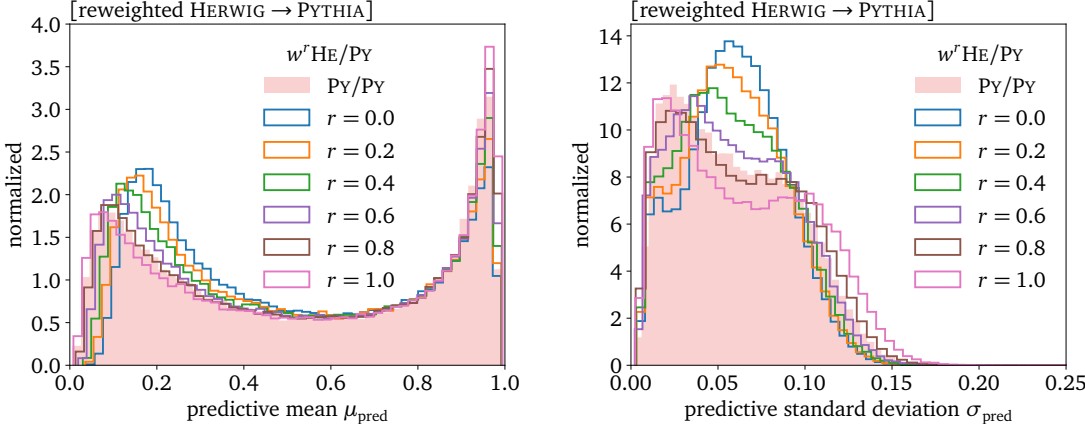

Figure 9: Bayesian ParticleNet-Lite, trained on reweighted HERWIG → PYTHIA jets and tested on PYTHIA jets. The curves should be compared to those in Fig. 6.

and does not allow us to claim that a general dependence on different training datasets is significantly reduced [6].

In the case of HERWIG vs. PYTHIA training for quark–gluon tagging the situation would be even worse, because the two datasets are systematically different in a way that is fully correlated with the features used for tagging. Decorrelating the difference of the two datasets would effectively remove $n_{\mathrm{PF}}$ from the available features and render the tagger useless. Instead, we need to find a way to best train the tagger and assign an uncertainty to this choice of training data.

**Interpolated training samples**

To add some resilience to the otherwise extreme choice of training either on HERWIG or on PYTHIA, we would like to use a combination of the two datasets for a stable training, benchmarked on the independent SHERPA data. There are, at least, two ways to interpolate between the two training datasets. First, we simply train the network on mixtures of quarks from PYTHIA and HERWIG vs. mixtures of gluons from PYTHIA and HERWIG in the same proportions,

$$\text{HERWIG training} \quad \xleftrightarrow{0 \le r \le 1} \quad \text{PYTHIA training}. \tag{10}$$

The interpolation parameter $r$ for the mixed sample is the fraction of PYTHIA jets in the training dataset.

An alternative method to achieve the same interpolated training is to train a discriminator on PYTHIA vs. HERWIG quark and gluon jets and to re-weight the HERWIG jets to their PYTHIA counterparts. Since each jet now comes with a weight, this method is also only defined on jet samples. This method has the advantage that we can train the network conditional on the interpolation parameter $r = 0 \dots 1$, to stabilize the training. In our case, the discriminator between HERWIG and PYTHIA jets is the same BPN-Lite network used to tag quarks vs. gluons. We use the same settings as in Tab. 2 and the same loss function as in Eq. (8). The only difference it that for the HERWIG vs. PYTHIA case we use generator truth-labels instead of jet truth-labels. We train the HERWIG vs. PYTHIA discriminator for quarks and gluons separately.

Using the per-jet reweighting factors from the classification network,

$$w(x_i) = \frac{p_{\mathrm{PY}}(x_i)}{p_{\mathrm{HE}}(x_i)}, \tag{11}$$

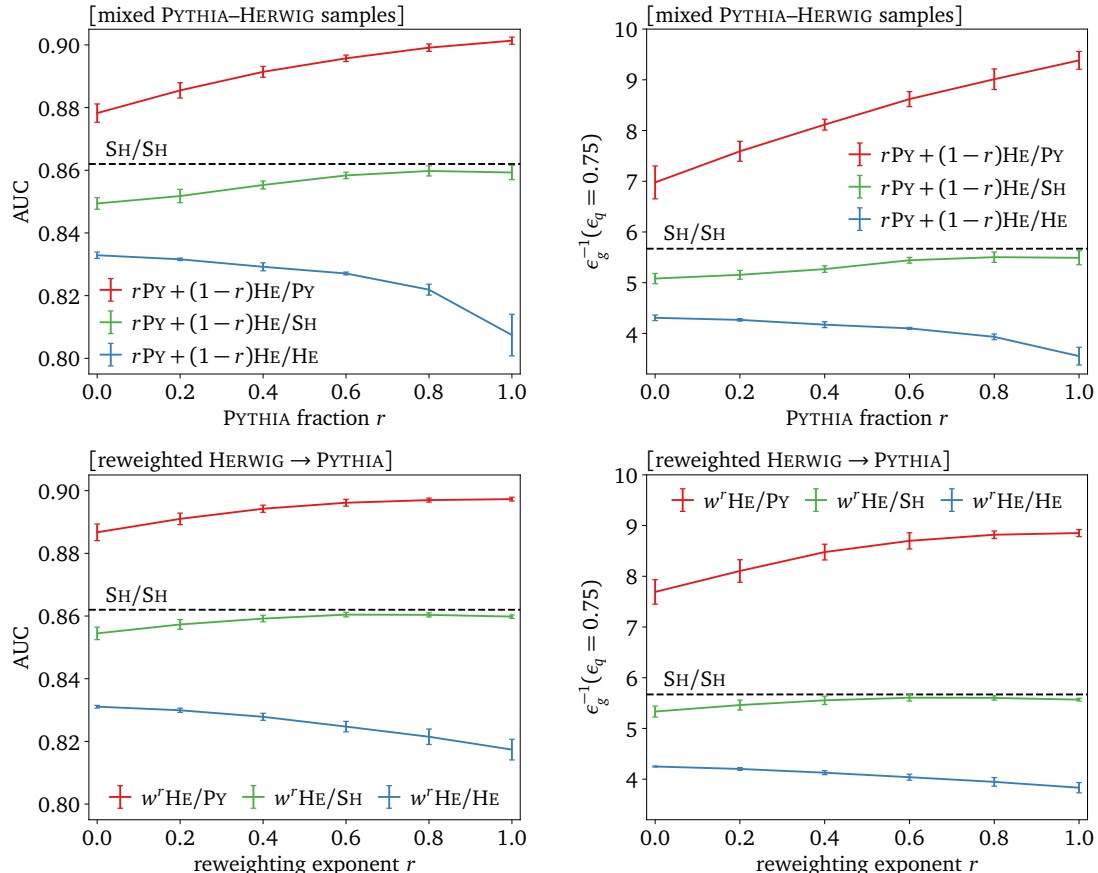

Figure 10: Performance of the interpolated training on HERWIG → PYTHIA, using mixed samples (*upper*) and conditional reweighting (*lower*). The performance is tested on pure HERWIG, PYTHIA, and independent SHERPA data. The error bars reflect six independent network trainings.

we can train a quark–gluon classifier on $w^r$-reweighted HERWIG jets. The weights enter the BPN-Lite loss function of Eq. (8) as

$$\mathcal{L}_{\text{BPN}} = -\frac{1}{M} \sum_{i=1}^{M} w(x_i)^r \log p(y_i|x_i, \omega) + \frac{1}{2N} \sum_{\text{weights } \omega_j} \left( \mu_j^2 + \sigma_j^2 - \log \sigma_j^2 - 1 \right), \qquad (12)$$

and the reweighting exponent $r$ is used as an additional feature input, uniformly sampled from $[0, 1]$ during training. In Fig. 9 we illustrate how the conditional reweighting network works on HERWIG jets. We show the distributions of the predictive mean $\mu_{\text{pred}}$ and the predictive uncertainty $\sigma_{\text{pred}}$ for a tagger trained conditionally on the weighted samples and tested on PYTHIA jets. In the limit $r \to 1$ the results approach the consistent PYTHIA training and testing shown in Fig. 6.

**Optimized training data and uncertainties**

The new aspect in this section is the performance of the interpolated training on the independent SHERPA data. Now, $r$ can be understood as a hyperparameter of the network training, so we can choose an optimal value from the independent calibration sample, in our case SHERPA. The actual tagging performance of the two methods of interpolated training is shown in Fig. 10, with mixed samples in the upper panels and reweighting in a conditional network setup in the

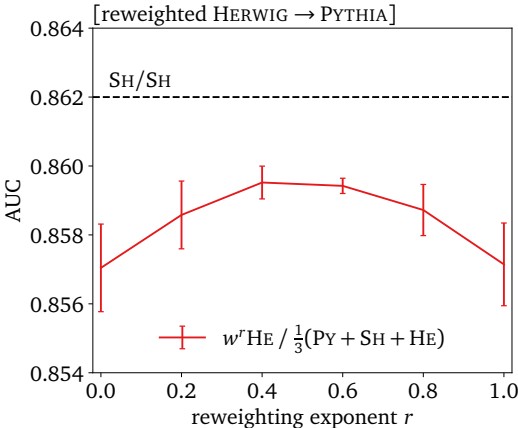
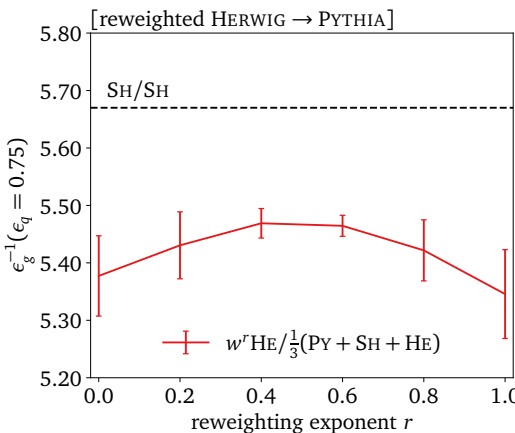

Figure 11: Performance of the interpolated training on HERWIG → PYTHIA, using conditional reweighting. The performance is tested on equal parts of HERWIG, PYTHIA, and SHERPA jets. The error bars reflect six independent network trainings.

lower panels. First, the results of the two methods are completely consistent with each other. They are also consistent with our previous results; for $r = 1$ the performance on all three test datasets approaches the results from proper PYTHIA training in Fig. 5. Similarly, in the limit $r = 0$ we reproduce the performance of HERWIG-based training. In between, we observe a continuous and featureless performance drop from PYTHIA to SHERPA training. Comparing the two methods, the conditional reweighting setup is smoother than the mixed sample training. Finally, after optimizing the interpolated training sample, the achieves performance comes very close to the consistent SHERPA training and testing.

As a side remark, while testing on SHERPA jets leads us to conclude that a choice $r \to 1$ provides the optimal tagging performance, we can also test the interpolated training on combination of HERWIG, PYTHIA, and SHERPA jets. Because the power of the main tagging features in the SHERPA dataset tends to lie in between HERWIG and PYTHIA, shown in Tab. 1, an interpolated training with $r \approx 0.5$ now gives the best tagging performance as shown in Fig. 11.

After optimizing the performance on a calibration dataset, we can also vary the interpolation parameter $r$ around its optimal value to estimate the uncertainty from our parameter choice. In the lower panels of Fig. 10 we see that for our setup the uncertainty from optimizing in the range $r \approx 0.5 \dots 1.0$ are significantly smaller than the variation from different network trainings. Strictly speaking, the performance gap even of the best training on the combined PYTHIA and HERWIG sample is significant, gauged by the uncertainty from the choice of $r$ and from different trainings. While our example interpolates between two samples, this kind of uncertainty estimate can easily be generalized to many training setups with a conditional reweighting network.

**Training-related, predictive uncertainties**

We can make use of the uncertainty-aware BPN-Lite tagger to provide the uncertainties $\sigma_{\text{pred}}$ for the interpolated training shown in Fig. 12. In analogy to the performance test in Fig. 10 we now show $\sigma_{\text{pred}}$ as a function of the interpolation parameter $r$. We know from Fig. 8 that the predictive uncertainties are given by the training data, and we can confirm that the interpolated training reproduces the small HERWIG uncertainties for $r = 0$ and the slightly larger PYTHIA uncertainties for $r = 1$. The reweighted and less consistent sample does not pose a challenge to the training, and the induced generalization errors are not large enough to affect the results for the different test datasets. As alluded to before, the interpolated training

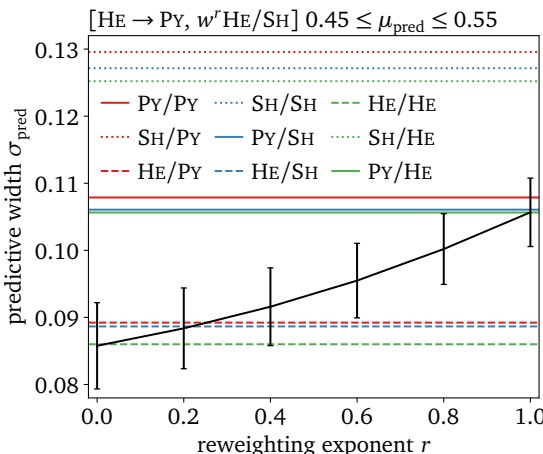

Figure 12: Predictive width for interpolated training on HERWIG → PYTHIA, using conditional reweighting. The error bars indicate the ranges from six independent trainings.

on PYTHIA and HERWIG comes with smaller uncertainties than consistent training on SHERPA, even when tested on SHERPA data. This can make sense, if the predictive uncertainties just reflect limitations in the training, for instance noise or stochasticity.

**Calibration and uncertainties**

One measurement where we expect the generalization error to appear is the calibration of the different taggers. In principle, the Bayesian PN-Lite tagger should be calibrated, but of course the calibration is only guaranteed when we train and test on consistent data. Any deviation from this consistency is expected to lead to a poorer calibration. In the left panel of Fig. 13 we first confirm that the consistent training and testing leads to well-calibrated taggers over the entire tagging score.

The picture changes when we train the tagger conditionally on the HERWIG–PYTHIA interpolation and evaluate the calibration on the independent SHERPA sample. In the right panel of Fig. 13 we see that HERWIG training leads to a well-calibrated tagger on the SHERPA dataset, reflecting the fact that the physics properties behind the two samples are similar. On the other hand, training on PYTHIA data leads to a poorly calibrated tagger on SHERPA data. Here, the fraction of correctly identified quark jets is lower than the score, which means the tagger is overconfident. This is consistent with PYTHIA being the dataset where it is easiest to separate quarks from gluons.

Because the change in the calibration curve reflects a more dramatic $r$-dependence than the network performance in Fig. 10 and the predictive uncertainty in Fig. 12, it provides the best handle on the generalization error which arises when we train a tagger flexibly on different generated samples and apply it to actual (calibration) data.

To summarize our findings from the interpolated training between HERWIG and PYTHIA and testing on SHERPA: if we are interested in the tagging performance only, we need to optimize $r \to 1$, corresponding to training on pure PYTHIA jets. When we want to minimize the Bayesian uncertainties from the training data, training with $r \to 0$, or on HERWIG, will give the smallest predictive uncertainties. Finally, when we want to maintain the tagger calibration, we again need to train on $r \to 0$ (HERWIG). Even in ML-applications there is no one size that fits all.

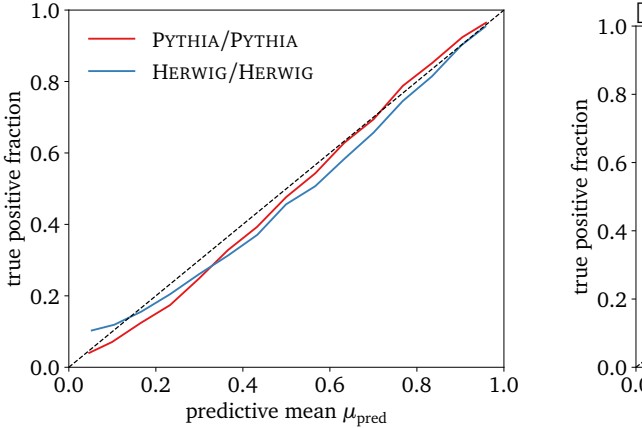 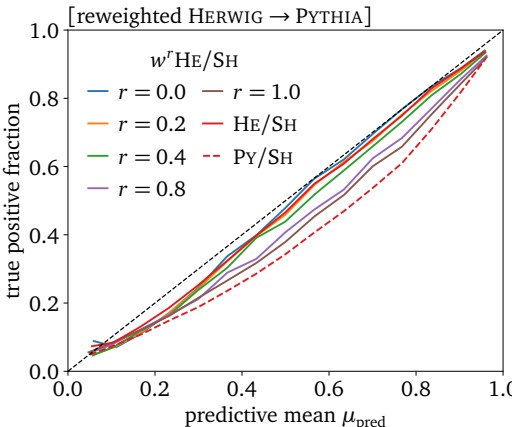

Figure 13: *Left:* calibration curves for consistent PYTHIA and HERWIG training and testing. *Right:* the same calibration curves using conditional HERWIG → PYTHIA reweighting on the training data, tested on SHERPA. We also show the benchmark results from training on pure HERWIG and PYTHIA jets, corresponding to $r \in \{0, 1\}$.

**Continuous calibration**

If it possible to train a tagger on a continuous interpolation between different datasets, the same kind interpolation should be possible between a simulated training dataset and an approximately labeled calibration dataset. A reliable training dataset should then be transferable into the actual data continuously, and without major changes in the performance and the behavior of the ML-tagger.

Instead of the interpolated training of Eq. (10), we now look at a triangle, defined by flexible training on an interpolated HERWIG–PYTHIA dataset and an additional interpolation between the training data and the independent SHERPA data,

$$\text{HERWIG/PYTHIA training} \quad \overset{0 \le r \le 1}{\longleftrightarrow} \quad \text{SHERPA training}. \tag{13}$$

In this scheme, we can interpolate several ways, within the training data, as described before, and from any kind of training data to the calibration data. The calibration data will not be properly labeled, so we can either rely on an approximate labeling or apply classification without labels [51]. The only tool required to implement this complex interpolation program is the same ParticleNet classification network that is used for the actual tagging task.

In Fig. 14 we illustrate the network training on a continuous interpolation between the simulated data and the calibration data. The benchmark performance is defined by the consistent SHERPA training and testing. As expected, there is hardly any change in performance when we train on PYTHIA jets, which means that the calibration of the tagger can focus on the correct calibration. The situation is different for HERWIG training data, where the performance indicates a generalization gap, but the calibration of the tagger is stable across the interpolation parameter. A situation like the one shown in Fig. 14 indicates that training the tagger on simulations (PYTHIA) and applying it to data is, essentially, optimal, with a very small uncertainty due to the generalization.

## 5 Conclusions

Now that it has become clear that ML-jet taggers will provide a transformative performance boost to a huge number of LHC analyses, we should turn our attention to aspects like resilience,

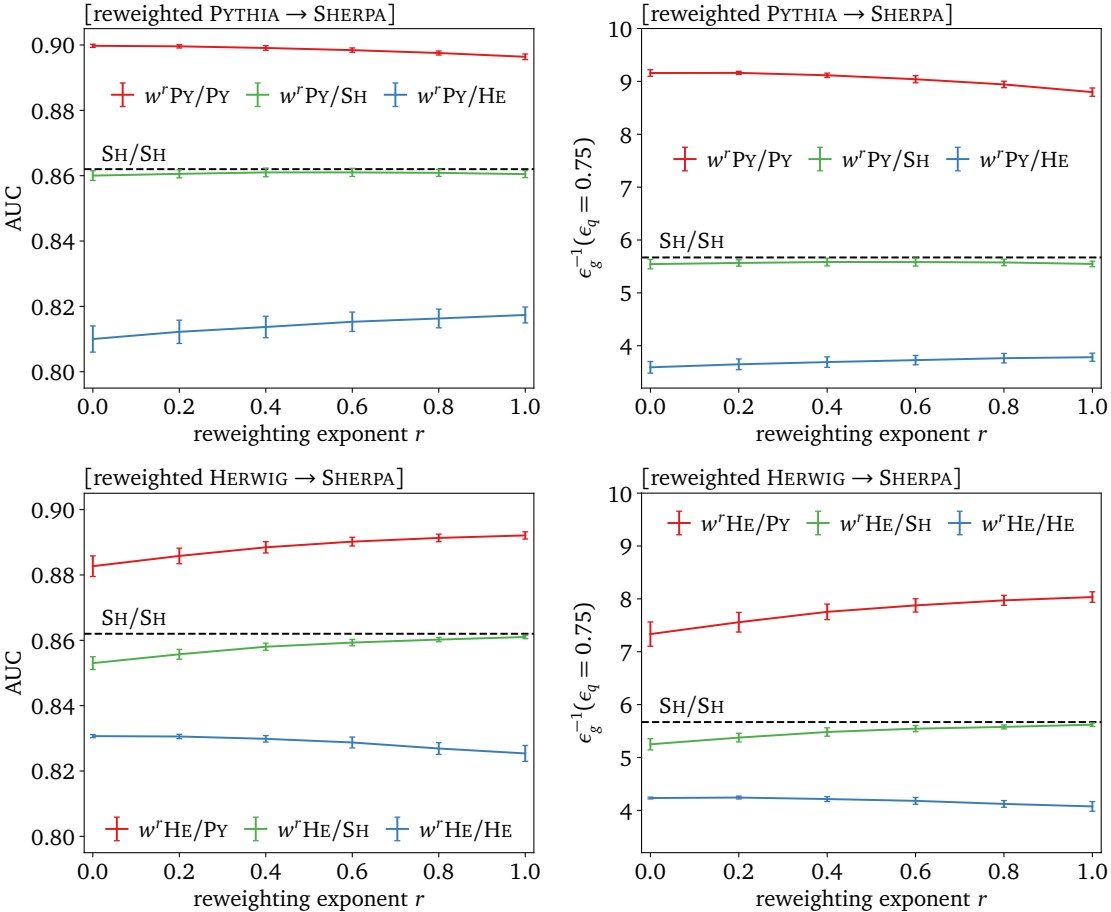

Figure 14: Performance of the interpolation between training and test data, using conditional reweighting PYTHIA → SHERPA (*upper*) and HERWIG → SHERPA (*lower*). The performance is benchmarked on pure HERWIG, PYTHIA, and SHERPA data. The error bars reflect six independent trainings.

uncertainties, control, or explainability. None of these are particularly strong points for classic multivariate taggers, so we again expect ML-taggers to further outperform traditional methods.

As long as we train taggers on simulations and test them on, or apply them to, an independent dataset, generalization errors will limit their performance, even if we remove biases through calibration. These generalization errors contribute to the theory uncertainty, specifically the dependence of the analysis outcome on the Monte Carlo simulation.

First, we have shown for quark–gluon tagging based on HERWIG and PYTHIA training data that improving the resilience through adversarial training is bound to fail, because the number of constituents is not only the leading tagging feature, it is also the main difference between the two simulations.

Just relying on two discrete datasets makes it hard to properly evaluate the corresponding theory uncertainty. We proposed a conditional training on a continuous interpolation between two training datasets, where the interpolation is best implemented using re-weighting through a classification network. The continuous interpolation parameter allows us to optimize the tagging performance and to estimate the related uncertainty. Our method can be generalized to larger numbers of training datasets and to continuous parameters describing the training datasets.

A Bayesian version of the ParticleNet(-Lite) subjet tagger allows us to track the stability of the conditional training and identify training-related uncertainties or even a breakdown of the interpolated training. For our application to quark–gluon tagging, trained on an interpolation between HERWIG and PYTHIA jets and tested on SHERPA jets, we find that from a pure performance perspective, training on PYTHIA gives the best results. They are very close to training on SHERPA and indicate a very small generalization gap. In contrast, if we are interested in small predictive uncertainties from the Bayesian network, we best train on HERWIG data. Similarly, for in a stable calibration HERWIG training also outperforms PYTHIA training, reflecting a common physics picture between HERWIG and SHERPA. For a test dataset combining the three generators, an interpolated training dataset right in between HERWIG and PYTHIA performs best. This indicates that different objectives require a flexible approach to simulation-based training.

Finally, we have speculated that our continuous interpolation between training samples can be generalized to an interpolation between training and calibration data, turning the actual calibration into a continuous procedure, where stability issues should be easily detectable.

## Acknowledgments

We would like to thank Michel Luchmann for inspiring discussions on Bayesian networks, Huilin Qu for expert advice on jet tagging, Theo Heimel for help with the SHERPA dataset, and Mathias Trabs for very helpful advice. A coordinated publication by Markus Klute and his group, reflecting inspiring discussions, will appear early in 2023. Furthermore, we would like to thank Fabrizio Klassen for his collaboration during an early phase of this project.

**Funding information**     AB and TP would like to thank the Baden-Württemberg-Stiftung for financing through the program *Internationale Spitzenforschung*, project *Uncertainties – Teaching AI its Limits* (BWST_IF2020-010). BMD acknowledges funding from the Alexander von Humboldt Foundation. This research is supported by the Deutsche Forschungsgemeinschaft (DFG, German Research Foundation) under grant 396021762 – TRR 257: *Particle Physics Phenomenology after the Higgs Discovery* and through Germany's Excellence Strategy EXC 2181/1 – 390900948 (the *Heidelberg STRUCTURES Excellence Cluster*).

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
