# Peer review of "Performance versus Resilience in Modern Quark-Gluon Tagging"

_SciPost Physics, doi:SciPost Phys. Core 6, 085 (2023)_

## Round 1 · Referee Report · Anonymous · 2023-4-19

Strengths
1 - Contribution to the question of quark and gluon tagging that is important
for a variety of LHC analyses.
2 - Focus on understanding and evaluating machine learning model and
uncertainties.
Weaknesses
1 - Some potentially confusing language, see below.
Report
The paper considers the tagging of QCD jets as quark or gluon like, with a focus
on the effect incompatible simulated data sets have on machine learning outcomes
and uncertainties. The study considers samples from the generators Pythia,
Herwig and Sherpa, whose differing description of quark and gluon jets are a
well known problem slowing down efforts in the community. The proposed solution
involves training on interpolated samples with a Bayesian network.
I think it is a very nice study that addresses the problem of dealing with
multiple, incompatible training inputs at the relevant example of inputs from
various generators to quark-gluon tagging. It is presented well and with
sufficient documentation. As such it can be published close to its current form
in my opinion. I have some minor comments below I would like to be addressed
first however.
Requested changes
1 - Strategies like training on Pythia or Herwig, or something in between, are
often referred to as 'good', 'optimal' or 'leading to smaller uncertainties' (or
the respective opposites), without qualification. I find this language somewhat
troublesome without context. As far as I understand, any of these statements are
relying on declaring some data set (for example Sherpa) as the 'correct' one? I
feel this should be clarified better, in particular in the conclusion that might
be read without detailed context.
2 - For example in Figure 6, but also others, I find the labels somewhat
confusing. The first generator is the training data set, and the second one was
tested on, right? It seems to me that is not explained anywhere.
3 - Figs. 10/11 state 'The error bars reflect six independent network
trainings.' What exactly does that mean? I also do not understand the
conclusion in the second to last paragraph on p. 15. In the lower panel of
Fig. 10, is the variation between 0.5 and 1 not larger than the error bars?
4 - The problem of identifying quark and gluon jets is described as
fundamentally 'undefined' at higher orders. However there is at least one
established algorithm, to define jet flavour at all orders (and some more as a
result of recent work). Would this potentially be helpful to repeat a study
like this with a higher order matched Monte Carlo?
Minor:
5 - On p. 10 in point 3, should it be $\mu_\text{pred}$ instead of $\sigma_\text{pred}$?
6 - Some figures appear to be out of order. Why is Fig. 1 included so
early but not described? Fig. 11 as far as I can tell is no mentioned anywhere
in the text.

---

## Round 1 · Referee Report · Anonymous · 2023-4-25

Strengths
1. Addresses an important problem in collider physics - tagging gluon-like and quark-like hadronic jets.
2. Outlines and demonstrates by way of a case study, techniques for using different, incompatible, simulated datasets to train a tagger, evaluate its performance, and estimate the associated uncertainty and bias.
Weaknesses
1. If I understood correctly, the underlying assumption is that the two training sets (Herwig and Pythia) span the theory uncertainty space and that the tagger performance may be understood/characterised by interpolating between them in training. There is no discussion of how this assumption might be tested in real data, and therefore it is not easy to see how a measurement uncertainty could be really defined. (If I have misunderstood please clarify!)
2. The simulated data sets are both implementations of QCD, along with phenomenological models for soft physics. While the "explainability" of ML taggers is mentioned as an advantage, there is no discussion of why these predictions differ so much, which seems like a missed opportunity - although arguably beyond the scope of the intended study.
Report
Given the strengths, this is definitely a publishable study, and meets the essential criteria.
Given the weaknesses, I think it might be better suited to core, since I think it presents incremental incremental progress in a long-standing problem, rather than a breakthrough.
Requested changes
Please add some citation(s) for the statement "ATLAS and CMS follow a more conservative approach and train taggers on simulations".
In 2.1 the authors say that c and b jets are not included. Does this refer to net flavour or are g-->ccbar, g--> bbbar splitting also excluded... please clarify in the text.
"all constituents have azimuthal angles φ within π of the jet.". I think for this algorithm they would be within delta-R of 0.4, so this statement looks redundant to me?
What does "one-hot encoding" mean? Is it a typo for "one-shot"? Even if so I'm not really sure...
When the authors say (near the end of section 2) the training on herwig supersedes that are pythia, this implies some time sequence. But if I understand rightly they just mean the training on herwig outperforms the training on pythia in this case? Please clarify in the text.
Not absolutely required, but if it is easy to do it would be would be clearer to show the herwig and pythia equivalent slices on the same plots, in Fig.7, since these are what you compare (rather than showing all herwig on one plot and all pythia on the other). I found it quite hard to judge the real similarities and differences discussed.

---

## Editorial Decision

published